# Context-Aware Clustering using Large Language Models

## Abstract

Despite the remarkable success of Large Language Models (LLMs) in text understanding and generation, their potential for text clustering tasks remains underexplored. While we observed powerful closed-source LLMs generate high-quality text clusterings, their massive size and inference cost make them impractical for repeated online use in real-world applications. Motivated by this limitation, we study the transfer of clustering knowledge from LLMs to smaller and more efficient open-source language models (SLMs), aiming to retain performance while improving scalability. We propose CACTUS (Context-Aware ClusTering with aUgmented triplet losS), a systematic approach that leverages SLMs for efficient and effective supervised clustering of entity subsets, particularly focusing on text-based entities. Existing text clustering methods fail to capture the context provided by the entity subset. In particular, they typically embed each entity independently, ignoring the mutual relationships among entities within the same subset. CACTUS incorporates a scalable inter-entity attention mechanism that efficiently models pairwise interactions to capture this context. Although several language modeling-based approaches exist for clustering, very few are designed for the task of supervised clustering. We propose a *new augmented triplet loss function* tailored for supervised clustering, which addresses the inherent challenges of directly applying the standard triplet loss to this problem by introducing a neutral similarity anchor. Furthermore, we introduce a *self-supervised clustering pretraining task* based on text augmentation techniques to improve the generalization of our model. Extensive experiments on various e-commerce query and product clustering datasets demonstrate that our proposed approach significantly outperforms existing unsupervised and supervised baselines across multiple external clustering evaluation metrics. *Our results establish CACTUS as a scalable, generalizable solution for real-world clustering scenarios.* Our code is publicly available at `https://anonymous.4open.science/r/context-aware-clustering-E90C`.

## 1 Introduction

Large Language Models (LLMs) have demonstrated human-level performance in text understanding and generation, but their application to text clustering tasks is underexplored. We observed that powerful closed-source LLMs (such as GPT-4 (Achiam et al., 2023) and Claude (Anthropic, 2023)), known for their instruction-following abilities, can provide high-quality clusterings through prompting. However, their massive size, high latency, and inference cost make them infeasible for repeated online use in real-world applications that demand low-latency and large-scale deployment. To overcome this limitation, we aim to develop a scalable model based on an open-source LM that can efficiently and effectively perform the clustering task. We study the problem of transferring the knowledge of clustering task from a powerful closed-source LLM to a scalable open-source SLM under the framework of supervised clustering, where the goal is to learn to cluster unseen entity[1] subsets, given training data comprising several examples of entity subsets with complete clusterings[2] (See Figure 1).

In this work, we focus particularly on entities described by short texts. The problem of clustering such entity subsets has applications in various domains including e-commerce, news clustering, and email management, among others (Finley & Joachims, 2005; 2008; Haider et al., 2007). However, deep learning approaches for

---

[1]In this paper, an entity refers to an individual concept or an item described by a short text, such as a search query or a product title.

[2]Complete clustering of a set refers to a clustering in which every entity in the set is assigned to a cluster.

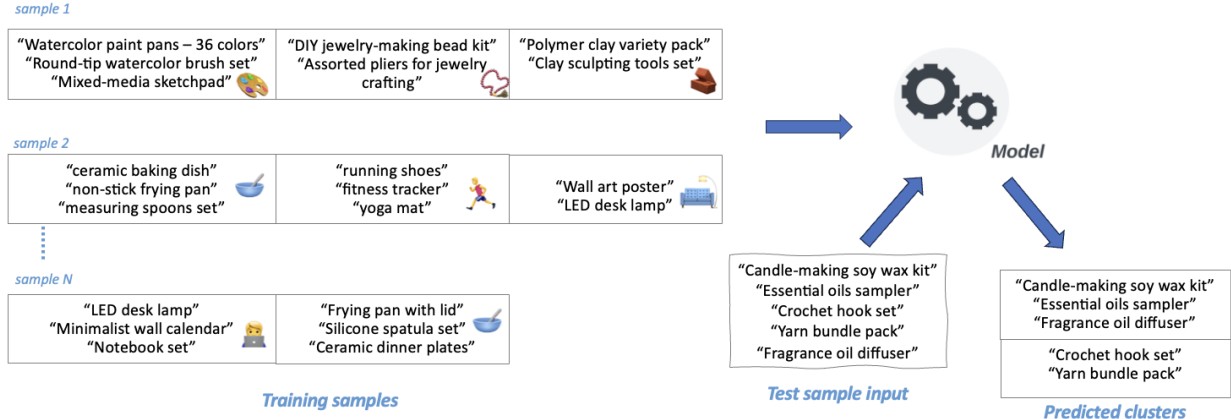

Figure 1: Illustration of the supervised clustering problem: Each training sample contains a subset of entities along with the corresponding ground truth clustering. Given a test input comprising an unseen entity subset, the goal is to cluster the entities in this subset.

solving the supervised clustering problem remain largely unexplored. Existing methods overlook the specific context provided by an entity subset and often rely on the latent structural loss function (Fernandes et al., 2012) which involves the sequential computation of maximum spanning forests. In our work, we propose an SLM-based solution called CACTUS (Context-Aware ClusTering with aUgmented triplet losS) that captures contextual information via inter-entity relationships, introduces an enhanced loss function, and incorporates a self-supervised clustering task.

The context of an entity subset refers to the unique circumstances that relate the specific entities co-occurring in the subset. This co-occurrence offers additional information influencing the interpretation and clustering of each entity. The context encompasses the relationships and interactions among entities in a subset, which helps in identifying shared themes or purposes. For example, consider the task of clustering a user's monthly purchases. A purchase of 'magnetic tape' could signify various intentions, such as for a science project or picture mounting. Examining the user's other purchases could provide the necessary context to help us determine the use case and place the entity in the appropriate cluster. However, most existing text clustering methods obtain a single embedding for each entity using a language model (Ahmed et al., 2022; Barnabo et al., 2023), thus ignoring the context. In contrast, our model computes entity embeddings that are dependent on the context or entity subset, which allows the model to identify entities with shared themes within the subset. The proposed method inputs the entire entity subset to the SLM and captures inter-entity interactions using a scalable attention mechanism, as traditional full attention over all entities in a subset can become computationally expensive as subsets grow large. Specifically, in each Transformer layer, for each entity, we compute a single representative embedding that participates in inter-entity attention.

Previous methods for supervised clustering applied the latent structural loss to pairwise entity features that are either hand-crafted or obtained from a neural network. While the latent structural loss involves sequential computations of spanning forests, the triplet loss can be parallelized (by processing all triplets in parallel using more memory) but faces the challenge of different triplets potentially having non-overlapping margin positions (see Section 3.3). To address this issue, we augment the complete graph of entities with a neutral entity, which is connected to all other entities by a learnable similarity score that provides a reference for all margin locations. Additionally, to further improve supervised clustering performance, especially in cases where ground truth clusterings are limited, we introduce a novel self-supervised clustering task. This task involves randomly sampling seed entities and constructing clusters with different transformations of each seed. This idea is inspired by text data augmentation techniques (Shorten et al., 2021) used in NLP tasks, but we formulate it, for the first time, as a self-supervised clustering task that aligns better with our finetuning phase.

In summary, the main contributions of our work are as follows:

- We propose a *new supervised clustering framework* for entity subsets that leverages context-aware entity embeddings derived from an SLM, enhanced with a scalable inter-entity attention (SIA) mechanism. This design enables efficient modeling of contextual relationships within subsets, even for large-scale datasets.

- We identify a key limitation in directly applying standard triplet loss to supervised clustering, namely, the misalignment of margin locations across triplets, and introduce a *novel augmented triplet loss function* that incorporates a learnable neutral similarity score to provide a consistent reference point across all triplets.

- We design a *dataset-specific self-supervised clustering pretraining task*, based on text augmentation techniques, to improve finetuning performance, particularly in low-resource settings where ground truth clusterings are scarce.

- Our extensive experiments demonstrate that CACTUS consistently outperforms both unsupervised and supervised state-of-the-art baselines across multiple real-world e-commerce query and product clustering datasets. We further validate our approach through comprehensive ablation studies, confirming each proposed component's individual and combined effectiveness.

## 2 Related Work

### 2.1 Traditional methods for supervised text clustering

The supervised clustering problem can be formulated as a binary pairwise classification task of predicting if a pair of entities belong to the same cluster. But this approach suffers from the drawback that the pairs are assumed to be i.i.d. (Finley & Joachims, 2005). Thus, structured prediction approaches have been explored as solutions to this problem. Traditional methods used hand-engineered pairwise features as inputs, where each pair of entities is described by a vector. Methods such as structural SVM (Tsochantaridis et al., 2004; Finley & Joachims, 2005) and structured perceptron (Collins, 2002) have been applied to this problem, where a parameterized scoring function is learned such that it assigns higher scores to correct clusterings in the training data. The scoring function depends on the pairwise features and the predicted clustering, and is formulated using correlation clustering (Bansal et al., 2002) or k-means (Finley & Joachims, 2008) frameworks. Observing that many within-cluster entity pairs have weak signals, Yu & Joachims (2009); Fernandes et al. (2012); Haponchyk et al. (2018) introduced maximum spanning forests over complete graphs of entities as latent structures in the scoring function. The inference stage involves finding a clustering with the highest score for a given entity subset. While traditional methods have relied on hand-engineered features, transformer-based language models which provide superior text embeddings create new opportunities for enhancing supervised text clustering.

### 2.2 Language models for text clustering

Despite the widespread use of Language Models (LMs) across diverse domains and applications, their application to 'supervised' clustering remains limited. Haponchyk & Moschitti (2021) and Barnabo et al. (2023) utilize encoder-only LMs to obtain pairwise and individual entity representations, respectively, and finetune the LMs using latent structural loss. The former is not a scalable approach as each entity pair is passed separately through a conventional Transformer model. In contrast to these existing methods, we propose a novel approach that passes the entire entity subset to a language model, and efficiently models inter-entity interactions within the Transformer layers, thereby improving clustering performance by capturing the unique context given by an entity subset. Furthermore, we depart from the latent structural loss (used in these existing works) that involves the sequential step of computing maximum spanning forests and employ an augmented triplet loss function that can be more easily parallelized and also achieves better performance.

It is worth noting that LMs have been widely applied to slightly different but more prevalent problems of unsupervised (Grootendorst, 2022; Zhang et al., 2021a;c; Meng et al., 2022) and semi-supervised clustering (Zhang et al., 2021b; Lin et al., 2020; Zhang et al., 2022; An et al., 2023). These tasks involve clustering of a single large entity set, with some pairwise constraints provided for semi-supervised clustering. Some recent works Viswanathan et al. (2023); Zhang et al. (2023); Nakshatri et al. (2023) take advantage of the

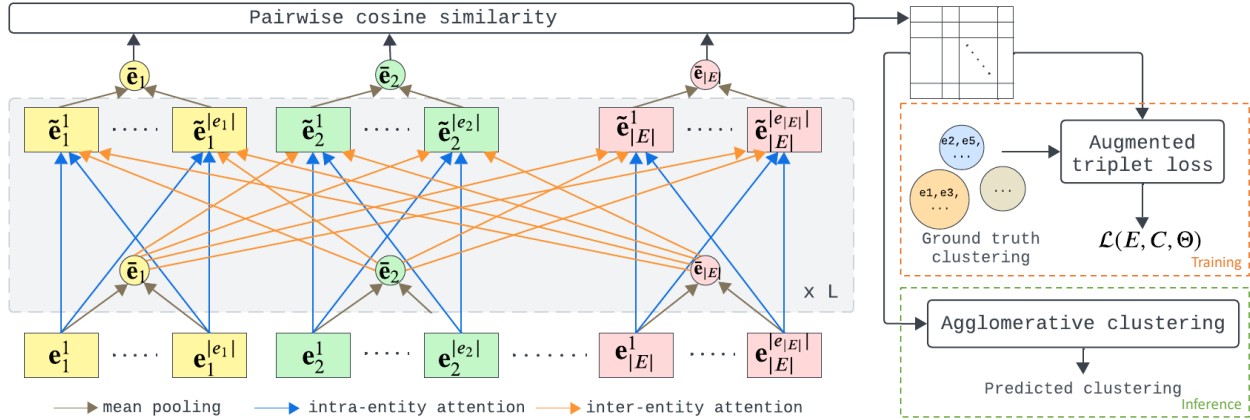

Figure 2: Overview of CACTUS: The entities in the input subset are tokenized and passed through an SLM, where the self-attention layers are modified with scalable inter-entity attention (SIA) to obtain context-aware entity embeddings. Pairwise cosine similarities are used for computing loss and predicted clusterings.

latest advances in LLMs by using them as oracles to make key decisions during the clustering process. However, these approaches are not suitable for our problem of clustering several entity subsets, as they require a new optimization problem for every new entity subset. Different from these LLM-based methods, our approach involves prompting an LLM to gather complete clusterings of several small entity subsets, which are subsequently used to fine-tune a scalable SLM that is adapted to capture the underlying context efficiently.

# 3 Proposed Method

This section provides a detailed description of the supervised clustering problem and our proposed method. Our approach involves finetuning an open-source pretrained Transformer encoder model for the task of context-aware clustering in a supervised manner. Here, 'context' refers to the subset in which an entity occurs, which influences the entity's interpretation. To capture context-awareness efficiently, we modify the self-attention layers of the SLM encoder to implement a scalable inter-entity attention mechanism, which is described in Section 3.2. We identify limitations of directly applying the triplet loss to supervised clustering and propose an augmented triplet loss function as a solution in Section 3.3. We further pretrain the SLM on a dataset-specific self-supervised clustering task before the finetuning phase, which is described in Appendix B due to space constraints. During inference, given an entity subset, we extract context-aware entity embeddings from the finetuned model, compute pairwise similarities, and feed them to an agglomerative clustering algorithm to obtain the predicted clustering. We chose agglomerative clustering for its ability to output different numbers of clusters with varying sizes for different entity subsets, given the same similarity/distance threshold. Alternatives to agglomerative clustering are discussed in Appendix D. We refer to the overall method as CACTUS (Context-Aware ClusTering with aUgmented triplet losS). Figure 2 provides an overview of the proposed approach.

## 3.1 Preliminaries

**Entity:** Let $\mathcal{E}$ be the universal set of entities in a dataset. An *entity* $e \in \mathcal{E}$ is an individual item, such as a search query or a product title, with a textual description denoted by $\texttt{text}(e)$.

**Clustering:** For an *entity subset* $E \subseteq \mathcal{E}$, a *clustering* $\mathcal{C} = (C, f)$ contains a set of clusters $C$ and an entity-to-cluster assignment function $f : E \twoheadrightarrow C$.[3] We say that two clusterings, $\mathcal{C} = (C, f)$ and $\mathcal{C}' = (C', f')$, over the same entity subset $E$ are equivalent if they induce the same partitioning of items i.e., if the

---

[3] '$\twoheadrightarrow$' denotes a surjective function.

pairwise co-cluster relationships are preserved. Formally, the clusterings $\mathcal{C}$ and $\mathcal{C}'$ are equivalent if and only if $\forall e_1, e_2 \in E,\ \ f(e_1) = f(e_2) \iff f'(e_1) = f'(e_2)$.

**Dataset:** A labeled dataset $\mathcal{D} = \{(E_1, \mathcal{C}_1), \ldots, (E_{|\mathcal{D}|}, \mathcal{C}_{|\mathcal{D}|})\}$ contains $|\mathcal{D}|$ samples where each sample contains an entity subset $E_k \subseteq \mathcal{E}$ and the corresponding ground truth clustering $\mathcal{C}_k$. We detail the process of collecting these clusterings using a closed-source model in Appendix A. These clusterings serve as ground truth in the dataset which is partitioned into training, validation, and test splits.

**Problem Statement (Supervised Clustering):** Given the training portion of the labeled dataset described above, the goal of supervised clustering is to learn a model that predicts a clustering for an unseen entity subset such that the predicted clustering is equivalent to the ground truth.

### 3.2 Context-awareness using Scalable Inter-entity Attention (SIA)

Here, we describe the architecture of the proposed Transformer encoder that is finetuned on the supervised clustering task using ground truth clusterings from an LLM. A common approach for text clustering involves obtaining a single embedding vector separately for each entity using a language model and defining a similarity or distance function in the embedding space, which is used in a clustering algorithm. We refer to this approach as NIA (No Inter-entity Attention) because there is no interaction between different entities in the embedding module. To capture context, i.e., to model entity embeddings that depend on the entity subset they occur in, we can also pass the entire subset in the input sequence and pool each entity's token embeddings. We refer to this approach as FIA (Full Inter-entity Attention), because all the token pairs among different entities are considered in the attention matrix. This is not very practical, especially, when entity descriptions are long and as the subsets grow large. So, we design a scalable inter-entity attention (SIA) mechanism that computes one representative embedding per entity which is used for inter-entity interactions. While scalable attention methods for handling long sequences in Transformers exist (Beltagy et al., 2020; Kitaev et al., 2020; Ainslie et al., 2020), this work is the first to explore scalable attention specifically in the context of clustering. The proposed SIA approach is described in detail below. We use the encoder of Flan-T5-base (Chung et al., 2022) as the underlying model and modify its attention layers for SIA.

Let $E = \{e_1, ..., e_{|E|}\}$ be an entity subset, where tokens of entity $e_i$ are denoted as $tokenize(text(e_i)) = (e_i^1, ..., e_i^{|e_i|})$. A Transformer-based LM gathers initial token embeddings and iteratively updates them using stacked Multi-Head Attention (MHA) and Feed Forward Network (FFN) layers. The Multi-Head Attention (MHA) layer traditionally computes all token-token pairwise attention scores, making it computationally intensive for long inputs. In SIA mechanism, we propose modifications to the MHA layer to make it more scalable for our clustering task. We split the attention computation into intra-entity and inter-entity components and make the latter more efficient by using pooled entity representations. Let $\mathbf{e_i^j} \in \mathbb{R}^d$ denote the embedding of token $e_i^j$ ($j^{th}$ token of $i^{th}$ entity) in the input to an MHA layer, and $\bar{\mathbf{e}}_\mathbf{i} = \frac{1}{|e_i|}\sum_k \mathbf{e_i^k}$ denote the mean-pooled representation of entity $e_i$. The MHA layer transforms the embedding $\mathbf{e_i^j}$ to $\tilde{\mathbf{e}}_\mathbf{i}^\mathbf{j} \in \mathbf{R}^d$ as follows. For simplicity, we show the computations for a single attention head and skip the projection layer at the end of MHA.

$$\tilde{\mathbf{e}}_\mathbf{i}^\mathbf{j} = \underbrace{\sum_{k=1}^{|e_i|} \alpha_{intra}(e_i^j, e_i^k)\mathbf{W^V}\mathbf{e_i^k}}_{\text{intra-entity attention}} + \underbrace{\sum_{\substack{m=1 \\ m \neq i}}^{|E|} \alpha_{inter}(e_i^j, e_m)\mathbf{W^V}\bar{\mathbf{e}}_\mathbf{m}}_{\text{inter-entity attention}} \tag{1}$$

$$\alpha_{intra(inter)}(e_i^j, .) = \frac{exp(Att_{intra(inter)}(e_i^j, .))}{\left( \begin{array}{c} \sum_{k=1}^{|e_i|} exp(Att_{intra}(e_i^j, e_i^k)) + \\ \sum_{\substack{m=1 \\ m \neq i}}^{|E|} exp(Att_{inter}(e_i^j, e_m)) \end{array} \right)} \tag{2}$$

$$Att_{intra}(e_i^j, e_i^k) = (\mathbf{W^Q}\mathbf{e_i^j})^T(\mathbf{W^K}\mathbf{e_i^k}) + \phi(k - i) \tag{3}$$

$$Att_{inter}(e_i^j, e_m) = (\mathbf{W^Q}\mathbf{e_i^j})^T(\mathbf{W^K}\bar{\mathbf{e}}_\mathbf{m}) \tag{4}$$

where $\mathbf{W^Q}$, $\mathbf{W^K}$, $\mathbf{W^V}$ $\in \mathbb{R}^{d \times d}$ are the query, key, and value projection matrices, respectively. Eq. equation 1 shows that a token within one entity attends to aggregated representations of other entities rather than individual tokens within those entities. The traditional softmax computation is altered in eq. equation 2 to separate the intra and inter-entity terms. The intra-entity attention in eq. equation 3 includes a relative positional encoding term, denoted by $\phi(.)$, while the inter-entity attention in eq. equation 4 does not. This is because the order of tokens within an entity is relevant while the order of entities in a subset is irrelevant. The token embeddings from the last Transformer layer are mean-pooled entity-wise to obtain the context-aware entity embeddings.

**Complexity:** Considering a subset of $N$ entities where each entity contains $L$ tokens, and a fixed embedding dimension $d$, the computational complexity of self-attention in the NIA embedding method is $O(NL^2)$ because there are $NL$ tokens in the entity subset, and each token only attends to the $L$ tokens within the same entity. In contrast, using the FIA approach increases the complexity to $O(N^2L^2)$ as each token attends to all $NL$ tokens from all entities. SIA provides a compromise between these two methods; it has $O(NL(L+N))$ complexity because each token attends to the $L$ tokens within the same entity and to $N-1$ representative entity embeddings.

### 3.3 Augmented Triplet Loss

After obtaining context-aware entity embeddings, we compute cosine similarity between all entity pairs in the input subset:

$$\text{sim}(e_i, e_k) = \frac{\bar{\mathbf{e}}_\mathbf{i}^\top \bar{\mathbf{e}}_\mathbf{k}}{\|\bar{\mathbf{e}}_\mathbf{i}\|\|\bar{\mathbf{e}}_\mathbf{k}\|} \tag{5}$$

The similarities are used to obtain predicted clusterings using the average-link agglomerative clustering algorithm. For the loss function, using these pairwise similarities as edge weights, we can construct a fully connected graph where each entity is a node. Previous methods for supervised clustering employed structural loss, which uses a scoring function based on a maximum spanning forest of the fully connected graph. This uses Kruskal's MST algorithm, which sequentially adds edges to the spanning forest and leads to slower loss computation. In contrast, the triplet loss (Schroff et al., 2015), which was shown to be a competitive baseline in Barnabo et al. (2023), can be easily parallelized as each triplet can be processed independently. For each entity in the input subset, the triplet loss considers other entities within the same cluster as positives and the remaining entities as negatives. For an entity subset $E$ with ground truth clustering $\mathcal{C} = (C, f)$, the triplet loss is given by

$$\mathcal{L}(E, \mathcal{C}, \Theta) = \frac{1}{|T(\mathcal{C})|} \sum_{(e, e_p, e_n) \in T(\mathcal{C})} \left( \gamma - \text{sim}(e, e_p) + \text{sim}(e, e_n) \right)^+ \tag{6}$$

where $\Theta$ are the parameters of the context-aware entity embedding module, $\gamma$ is the margin which is a hyperparameter, and $T(\mathcal{C}) = \{(e, e_p, e_n) : e, e_p, e_n \in E; e \neq e_p; f(e) = f(e_p) \neq f(e_n)\}$ is the set of triplets.

The triplet loss formulation presents a challenge due to potential non-overlapping margin locations across different triplets. Margin location refers to the range between similarities from anchor entity ($e$) to positive ($e_p$) and negative ($e_n$) entities within a triplet. For example, in Figure 3 with three clusters containing two entities each, the pairwise similarities shown result in the minimum value for triplet loss. However, there exist inter-cluster edges with higher similarity than an intra-cluster edge, which results in 'green' and 'blue' clusters being merged by the

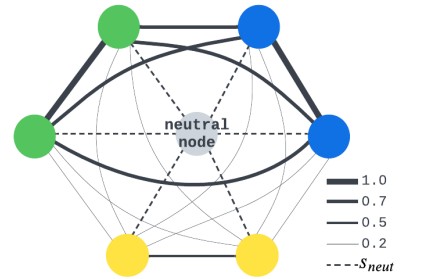

Figure 3: Example of an entity subset with 3 clusters containing 2 entities each. There exists an intra-cluster (yellow) edge with similarity less than some inter-cluster (green-blue) edges. For margin=0.3, the triplet loss (eq. 6) is at its minimum while the proposed augmented triplet loss (eq. 7) is not.

agglomerative clustering algorithm before the 'yellow' cluster is formed. This phenomenon can also occur for intra and inter-cluster edges in different entity subsets, which makes it difficult to choose a global threshold for agglomerative clustering during inference. To avoid such problems, we augment the complete graph with a neutral node that is connected to all other entities via a learnable neutral edge similarity $s_{neu}$. The neutral node is incorporated into the augmented triplet loss to encourage intra and inter-cluster edge similarities to lie on opposite sides of $s_{neu}$. We do not learn the neutral bode embedding explicitly but only learn the neutral edge similarity. Not using a global neutral entity embedding allows the flexibility of having different implicit neutral entity embeddings for each entity subset, though we assume $s_{neu}$ to be the same globally. The new loss function is given by

$$\mathcal{L}^{aug}(E, \mathcal{C}, \Theta) = \frac{1}{T'(C)} \Bigg\{ \sum_{(e,e_p,e_n) \in T(\mathcal{C})} (\gamma - \text{sim}(e, e_p) + \text{sim}(e, e_n))^+$$

$$+ \sum_{(e,e_p) \in P^{intra}(\mathcal{C})} (\frac{\gamma}{2} - \text{sim}(e, e_p) + s_{neu})^+$$

$$+ \sum_{(e,e_n) \in P^{inter}(\mathcal{C})} (\frac{\gamma}{2} - s_{neu} + \text{sim}(e, e_n))^+ \Bigg\} \tag{7}$$

$$T'(C) = |T(\mathcal{C})| + |P^{intra}(\mathcal{C})| + |P^{inter}(\mathcal{C})| \tag{8}$$

where $P^{intra}(\mathcal{C}) = \{(e, e_p) : (e, e_p, .) \in T(\mathcal{C})\}$ is the set of entity pairs within the same cluster and $P^{inter}(\mathcal{C}) = \{(e, e_n) : (e, ., e_n) \in T(\mathcal{C})\}$ is the set of entity pairs in different clusters. The newly added loss terms encourage the intra-cluster (inter-cluster) pairwise similarities to be $\frac{\gamma}{2}$ higher (lower) than the neutral edge similarity. Thus, the neutral edge similarity softly constraints the margin location for all triplets.

Table 1: Dataset statistics. (* Since the Gifts dataset is proprietary, we provide approximate numbers for the statistics reported.)

|  | Gifts* | Arts | Games | Instruments | Office |
|---|---|---|---|---|---|
| No. of entities | ∼365K | 22,595 | 16,746 | 10,522 | 27,532 |
| No. of entity subsets | ∼42K | 55,629 | 54,995 | 27,420 | 100,775 |
| Avg. size of entity subset | ∼46 | 5.4 | 5.7 | 5.6 | 5.0 |
| Avg. no. of clusters per entity subset | ∼6 | 2.6 | 2.8 | 2.8 | 2.7 |
| Avg. no. of entities per cluster | ∼8 | 2.1 | 2.1 | 2.0 | 1.9 |
| Avg. no. of words per entity | ∼3 | 11.6 | 6.9 | 10.5 | 13.9 |

## 4 Experiments

In this section, we describe the datasets utilized for our experiments and compare the proposed method to existing unsupervised and supervised clustering baselines using external clustering evaluation metrics. We also conduct ablation studies to analyze the effectiveness of the different components of our method. Additionally, we present a qualitative study to illustrate how context-awareness improves clustering performance. Cross-dataset generalization experiments are discussed in Appendix C.

### 4.1 Experimental Setup

We compile five datasets for our experiments, including four from Amazon product reviews (Ni et al., 2019) and one proprietary dataset called Gifts. The Amazon datasets, including Arts, Games, Instruments, and Office, consist of sequences of products reviewed by users, with each user's product sequence treated as one entity subset. These subsets can also be seen as purchase lists of individual users. The purpose of grouping these entity subsets is to organize users' purchase histories into meaningful clusters, enabling users to view their past purchases in well-defined groups. We use preprocessed datasets from Li et al. (2023), considering product titles as textual descriptions of entities.

The Gifts dataset contains search queries related to 'gifts' from an e-commerce platform. Each entity subset contains potential next search queries linked to a particular source query on the platform. Clustering these entity subsets organizes possible next search queries into meaningful clusters, helping users refine or edit their searches more effectively by presenting well-defined groups of relevant query suggestions.

Basic statistics of the datasets are summarized in Table 1. On average, the Amazon datasets contain 5 to 6 entities per entity subset, while Gifts contains approximately 46 entities. For each dataset, we randomly sample 3K entity subsets for test split, 1K subsets for validation split, and use the remaining for training.

We use a proprietary model to collect ground truth clusterings for all datasets. We perform self-supervised pretraining for the Amazon datasets but not for Gifts, as the queries in Gifts are very short, making it difficult to obtain multiple transformations of a query. We evaluate the predicted clusterings from the finetuned SLM (Flan-T5-base) by comparing them to the ground truth clusterings. We use the following extrinsic clustering evaluation metrics: Normalized Mutual Information (NMI), Adjusted Mutual Information (AMI) Vinh et al. (2009), Rand Index (RI), Adjusted Rand Index (ARI) Hubert & Arabie (1985), and F1-score Haponchyk et al. (2018).

### 4.2 Comparison with Baselines

To evaluate our model, we compare it with the following unsupervised and supervised clustering baselines:

- *Unsupervised*: As unsupervised clustering baselines, we employ the **K-Means**, **Spectral**, and **Agglomerative** clustering algorithms. The entity embeddings for unsupervised baselines are obtained from the pretrained Flan-T5-base encoder. For K-Means and Spectral clustering, we determine the number of clusters for each entity subset using either the silhouette method or the average number from the training set, based on validation metrics. For agglomerative clustering, we use cosine similarity with average linkage, with the threshold set according to the validation set.

- *Supervised*: The supervised baselines, described below, are trained using the ground truth clusterings, similar to the proposed method.
    - **Word-emb**: Each entity's embedding is computed as the average of embeddings for all tokens within that entity, without incorporating any context about the entity subset. The word embeddings are initialized using the same Flan-T5-base model used for CACTUS, and then finetuned.
    - **Word-emb (context)**: This baseline extends word-emb by concatenating each entity's embedding with the "average of entity embeddings in the subset". The concatenated representation is then fed to an MLP to generate a context-aware entity embedding.
    - **SCL** Barnabo et al. (2023): Each entity is processed independently through a Language Model to generate its embedding. The model is finetuned using the structural loss which computes maximum spanning forests.
    - **SCL (context)**: Similar to "word-emb (context)", this approach extends SCL by incorporating subset context. It concatenates averaged entity embeddings within a subset to each entity's embedding and then uses an MLP to produce a context-aware entity embedding.

For the supervised baselines, we did not include NSC (Haponchyk & Moschitti, 2021) as it demands substantial GPU memory and often leads to OOM errors. For a fair comparison, we employ FlanT5-base encoder as the SLM for all baselines. Table 2 shows the evaluation metrics for CACTUS and the baselines along with 95% confidence intervals. Among the unsupervised methods, agglomerative clustering performs best in most cases. The supervised methods achieve significantly better results than the unsupervised baselines, demonstrating that supervision is necessary to capture clustering patterns in these datasets. Both SCL(context) and word-emb (context) outperform SCL and word-emb respectively, which highlights the importance of capturing context in an entity subset. SCL also surpasses word-emb, indicating that the transformer layers provide better entity representations. CACTUS significantly outperforms all the unsupervised and supervised baselines, improving the AMI and ARI metrics over SCL (context) by 9.6%-18.2% and 8.7%-16.9%, respectively.

Table 2: Comparison of the proposed method to previous unsupervised and supervised clustering baselines. For each metric, we show the mean across all test samples along with 95% CI. The first three are unsupervised methods, and the next five are supervised clustering methods. *For the proprietary Gifts dataset, we report improvements against K-Means.

| | Model | NMI | AMI | RI | ARI | F1 |
|---|---|---|---|---|---|---|
| Gifts* | K-Means | +0.000±0.006 | +0.000±0.006 | +0.000±0.004 | +0.000±0.006 | +0.000±0.004 |
| | Spectral | +0.020±0.006 | +0.024±0.006 | -0.002±0.004 | +0.006±0.006 | +0.014±0.004 |
| | Agglomerative | +0.047±0.008 | +0.009±0.008 | -0.019±0.006 | +0.011±0.008 | +0.027±0.004 |
| | word-emb | +0.082±0.010 | +0.101±0.009 | -0.003±0.008 | +0.108±0.010 | +0.079±0.005 |
| | word-emb (context) | +0.114±0.007 | +0.175±0.008 | +0.047±0.005 | +0.185±0.008 | +0.109±0.004 |
| | SCL | +0.167±0.007 | +0.196±0.008 | +0.065±0.006 | +0.195±0.009 | +0.114±0.004 |
| | SCL (context) | +0.164±0.007 | +0.220±0.007 | +0.078±0.004 | +0.225±0.008 | +0.122±0.004 |
| | CACTUS (ours) | **+0.207**±0.006 | **+0.260**±0.007 | **+0.098**±0.004 | **+0.263**±0.008 | **+0.144**±0.004 |
| Arts | K-Means | 0.660±0.010 | 0.167±0.016 | 0.690±0.009 | 0.250±0.018 | 0.766±0.006 |
| | Spectral | 0.642±0.010 | 0.192±0.012 | 0.683±0.008 | 0.272±0.015 | 0.790±0.005 |
| | Agglomerative | 0.692±0.010 | 0.219±0.013 | 0.707±0.009 | 0.290±0.015 | 0.781±0.006 |
| | word-emb | 0.717±0.010 | 0.348±0.015 | 0.741±0.009 | 0.407±0.016 | 0.823±0.005 |
| | word-emb (context) | 0.716±0.010 | 0.384±0.016 | 0.760±0.009 | 0.461±0.016 | 0.847±0.005 |
| | SCL | 0.725±0.010 | 0.371±0.015 | 0.751±0.009 | 0.435±0.016 | 0.833±0.005 |
| | SCL (context) | 0.749±0.010 | 0.412±0.015 | 0.776±0.008 | 0.486±0.016 | 0.853±0.005 |
| | CACTUS (ours) | **0.764**±0.010 | **0.461**±0.016 | **0.795**±0.008 | **0.540**±0.016 | **0.868**±0.005 |
| Games | K-Means | 0.681±0.010 | 0.213±0.017 | 0.712±0.009 | 0.247±0.018 | 0.767±0.006 |
| | Spectral | 0.688±0.010 | 0.230±0.017 | 0.718±0.009 | 0.263±0.017 | 0.771±0.006 |
| | Agglomerative | 0.640±0.011 | 0.268±0.015 | 0.691±0.009 | 0.291±0.016 | 0.799±0.005 |
| | word-emb | 0.689±0.011 | 0.376±0.016 | 0.741±0.009 | 0.398±0.016 | 0.829±0.005 |
| | word-emb (context) | 0.730±0.010 | 0.425±0.016 | 0.770±0.008 | 0.454±0.016 | 0.843±0.005 |
| | SCL | 0.718±0.011 | 0.442±0.016 | 0.763±0.009 | 0.462±0.016 | 0.849±0.005 |
| | SCL (context) | 0.748±0.009 | 0.471±0.016 | 0.787±0.008 | 0.499±0.016 | 0.857±0.005 |
| | CACTUS (ours) | **0.777**±0.010 | **0.540**±0.016 | **0.813**±0.008 | **0.565**±0.016 | **0.876**±0.005 |
| Instruments | K-Means | 0.678±0.009 | 0.181±0.016 | 0.705±0.008 | 0.213±0.017 | 0.764±0.006 |
| | Spectral | 0.686±0.009 | 0.196±0.017 | 0.713±0.008 | 0.229±0.017 | 0.767±0.006 |
| | Agglomerative | 0.707±0.009 | 0.226±0.013 | 0.719±0.008 | 0.257±0.014 | 0.776±0.005 |
| | word-emb | 0.735±0.009 | 0.351±0.015 | 0.754±0.008 | 0.369±0.015 | 0.819±0.005 |
| | word-emb (context) | 0.758±0.009 | 0.448±0.016 | 0.789±0.008 | 0.478±0.016 | 0.852±0.005 |
| | SCL | 0.728±0.010 | 0.436±0.016 | 0.765±0.008 | 0.451±0.016 | 0.849±0.005 |
| | SCL (context) | 0.773±0.009 | 0.497±0.016 | 0.804±0.007 | 0.525±0.016 | 0.868±0.004 |
| | CACTUS (ours) | **0.786**±0.009 | **0.553**±0.016 | **0.817**±0.008 | **0.578**±0.016 | **0.883**±0.004 |
| Office | K-Means | 0.731±0.009 | 0.267±0.017 | 0.748±0.008 | 0.332±0.018 | 0.808±0.006 |
| | Spectral | 0.735±0.009 | 0.275±0.017 | 0.752±0.008 | 0.340±0.018 | 0.809±0.006 |
| | Agglomerative | 0.748±0.009 | 0.324±0.015 | 0.760±0.008 | 0.383±0.016 | 0.829±0.005 |
| | word-emb | 0.773±0.009 | 0.412±0.015 | 0.788±0.008 | 0.472±0.016 | 0.856±0.005 |
| | word-emb (context) | 0.790±0.009 | 0.477±0.016 | 0.814±0.008 | 0.544±0.016 | 0.878±0.004 |
| | SCL | 0.772±0.010 | 0.445±0.016 | 0.792±0.008 | 0.500±0.016 | 0.866±0.005 |
| | SCL (context) | 0.800±0.009 | 0.513±0.016 | 0.822±0.008 | 0.576±0.016 | 0.887±0.004 |
| | CACTUS (ours) | **0.821**±0.009 | **0.562**±0.016 | **0.842**±0.008 | **0.626**±0.016 | **0.902**±0.004 |

Table 3: Results on validation set using different architectures for entity subset encoder. Proposed method (Section 3.2) is indicated by *. Augmented triplet loss is used to train all models.

| | Set encoder | AMI | ARI | F1 |
|---|---|---|---|---|
| Arts | NIA | 0.354 | 0.409 | 0.826 |
| | NIA (context) | 0.354 | 0.419 | 0.829 |
| | SIA (KV-mean) | 0.398 | 0.450 | 0.840 |
| | SIA (first) | 0.396 | 0.461 | 0.841 |
| | SIA (hid-mean)* | 0.398 | 0.467 | 0.845 |
| | FIA | **0.423** | **0.494** | **0.851** |
| Office | NIA | 0.442 | 0.495 | 0.867 |
| | NIA (context) | 0.491 | 0.542 | 0.877 |
| | SIA (KV-mean) | 0.470 | 0.526 | 0.875 |
| | SIA (first) | 0.493 | 0.552 | 0.881 |
| | SIA (hid-mean)* | **0.513** | **0.568** | **0.885** |
| | FIA | 0.493 | 0.553 | 0.879 |

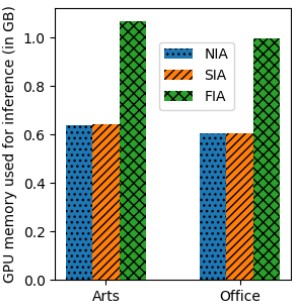 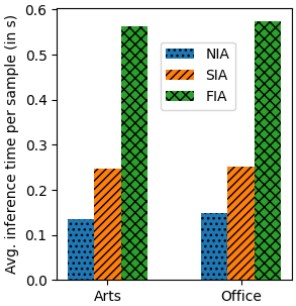

Figure 4: Comparison of GPU memory usage and average inference time per sample for NIA, SIA (hid-mean), and FIA. For computing memory usage (left), we use the same batch size of 16 for all methods. For measuring inference time (right), we set the batch size to 16 for FIA while the batch sizes for NIA and SIA are increased until memory usage for these methods does not exceed that of FIA.

### 4.3 Ablation Studies

We conduct ablation experiments to evaluate the effectiveness of the various proposed components, including context-aware entity embeddings, the augmented triplet loss function, and the self-supervised clustering task. For faster training, we use 3K training entity subsets instead of the full dataset for ablation studies. We focus on AMI, ARI, and F1 scores, omitting NMI and RI. The latter can sometimes be high for random clusterings and are not adjusted for chance, unlike AMI and ARI (Vinh et al., 2009).

**Set encoder.** We compare six different methods for obtaining entity embeddings; the results are shown in Table 3. The NIA, SIA(hid-mean), and FIA methods are described in section 3.2. We explore two more scalable attention mechanisms: SIA(KV-mean) where keys and values are pooled instead of the hidden representations, and SIA(first) where the first token in each entity is used as the representative token for inter-entity attention. We also consider NIA(context), which is similar to SCL(context) (see Section 4.2), but uses the proposed augmented triplet loss. NIA(context), SIA, and FIA methods obtain better results than NIA which demonstrates the importance of capturing context given by an entity subset. SIA(first) and SIA(hid-mean) outperform NIA(context) on both datasets, indicating that the inter-entity attention in the Transformer layer provides better embeddings than simply concatenating the average entity embedding to each entity embedding. The FIA method achieves the best results on the Arts dataset, while SIA (hid-mean) achieves the best results on the Office dataset. Among the three SIA methods, SIA (hid-mean) yields the highest metrics on both datasets.

Figure 4 (left) shows the usage of GPU memory during inference for NIA, SIA (hid-mean), and FIA embedding methods. Each batch consists of 16 entity subsets. Details of the batch processing are provided in Appendix E.1. Memory usage is measured in torch using "torch.cuda.memory.max_memory_allocated". SIA's memory usage is very similar to that of NIA but about 40% less compared to FIA. For all three methods, 0.44GB of GPU memory is used to hold the model with 110M parameters of the fp32 data type.

Since NIA and SIA consume less memory than FIA, we can increase their batch sizes to reduce inference time. Therefore, for measuring inference time, we increase the batch sizes for NIA and SIA methods until their memory usage does not exceed that of the FIA. This allows for a fair comparison of inference time by allowing all the methods to have the same memory limit. Figure 4 (right) shows the average inference time per sample for the three methods. We see approximately 56% reduction in inference time from FIA to SIA and about 41%-45% reduction from SIA to NIA.

Table 4: Results on the validation set using different supervised clustering loss functions for training. SIA (first) architecture is used for the set encoder.

|  | Loss | AMI | ARI | F1 |
|---|---|---|---|---|
| Arts | cross-entropy | 0.374 | 0.441 | 0.832 |
|  | structural loss | 0.385 | 0.441 | 0.835 |
|  | triplet | 0.389 | 0.444 | 0.837 |
|  | augmented triplet | **0.396** | **0.461** | **0.841** |
| Office | cross-entropy | 0.488 | 0.548 | 0.876 |
|  | structural loss | 0.494 | 0.549 | **0.881** |
|  | triplet | **0.497** | 0.543 | 0.880 |
|  | augmented triplet | 0.493 | **0.552** | **0.881** |

Table 5: Results on validation set with and without self-supervision. SIA (hid-mean) architecture is used for the set encoder. (SS: Self-supervision)

|  | SS | AMI | ARI | F1 |
|---|---|---|---|---|
| Arts | ✗ | 0.398 | 0.467 | 0.845 |
|  | ✓ | **0.446** | **0.502** | **0.855** |
| Office | ✗ | 0.513 | 0.568 | 0.885 |
|  | ✓ | **0.552** | **0.608** | **0.894** |

**Loss function.** We compare different loss functions to evaluate their impact on clustering performance. These include the triplet and the augmented triplet loss functions described in Section 3.3, as well as the structural loss Haponchyk & Moschitti (2021), and the binary cross-entropy loss typically used for pairwise classification. As shown in Table 4, the augmented triplet loss achieves the highest AMI, ARI, and F1 scores on the Arts dataset, and the highest ARI and F1 scores on the Office dataset.

**Self-supervision.** Table 5 compares the clustering performance of our model with and without the proposed self-supervised pretraining phase, which is described in Appendix B. In both cases, we initialize the model with pretrained FlanT5 weights, but for the self-supervised approach, we include an additional dataset-specific pretraining phase before fine-tuning. This self-supervised clustering task enhances the model's understanding of the clustering problem, leading to improved AMI, ARI, and F1 scores on both Arts and Office datasets.

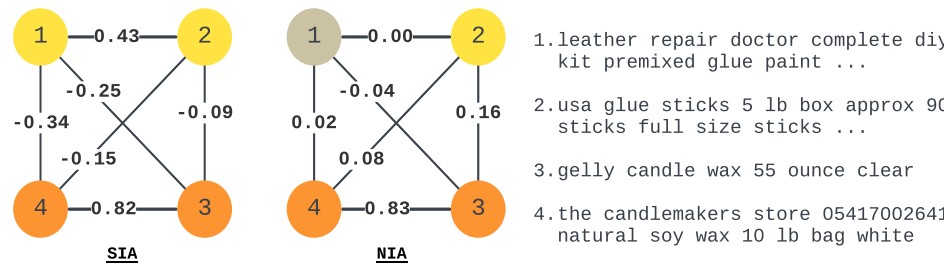

Figure 5: Case Study: Predicted clusterings with pairwise similarities using SIA and NIA methods. The SIA method correctly identifies the common cluster membership of the first two entities where NIA fails. The stopping threshold for agglomerative clustering is chosen based on the results of the validation set.

## 4.4 Qualitative Analysis

To illustrate the importance of context-aware embeddings, we provide a qualitative example. Referring to Figure 5, using SIA embeddings, our model accurately identifies two products each under the 'Glue Products' and 'Candle Making Supplies' clusters. However, with NIA embeddings, the model fails to capture the similarity between the two glue products. Specifically, in the NIA embeddings, the first product, a leather repair glue paint, is placed closer to other leather repair products in the universal entity set but far away from products containing glue sticks. The SIA approach leverages the context provided by the current entity subset and places the leather repair glue paint and glue sticks (the first two entities) within the same cluster.

# 5 Conclusion

This paper presented a novel approach for supervised clustering of entity subsets using context-aware entity embeddings from Transformer-based models. Context-awareness is achieved through a scalable inter-entity attention mechanism that facilitates interactions among different entities at each layer of the SLM. We proposed the augmented triplet loss to overcome the challenges associated with directly applying the traditional triplet loss to supervised clustering. Additionally, we introduced a self-supervised clustering task that enhances finetuning performance. By integrating the proposed components, we demonstrated that our model significantly outperforms existing methods across various extrinsic clustering evaluation metrics. Future research could investigate alternative techniques for inter-entity attention, explore additional loss functions and self-supervision tasks, and extend this work to other application domains.

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

# APPENDIX

## A   Collecting ground truth for Supervised Clustering

We collect ground truth clusterings for entity subsets by prompting a proprietary closed-source LLM. Given an entity subset $E = \{e_1, ..., e_{|E|}\}$, we use the following prompt as input to the model.

> Cluster these products:
> $\texttt{text}(e_1)$
> $\vdots$
> $\texttt{text}(e_{|E|})$
> For each cluster, the answer should contain a meaningful cluster title and the products in that cluster. Do not provide any explanation.

Based on our observation of the text outputs for a few cases, we implement a parsing algorithm to convert the LLM's text outputs into clusterings. We discard outputs that are either empty or cannot be parsed (fewer than 0.08% of entity subsets). Additionally, entities that do not appear in the parsed clusterings (less than 3% per sequence on average) are removed. The distributions of the number of clusters per entity subset for each of the five datasets described in Section 4.1 are shown in Figure A1. On average, the Gifts dataset contains 6 clusters and the Amazon datasets contain about 2 clusters per entity subset.

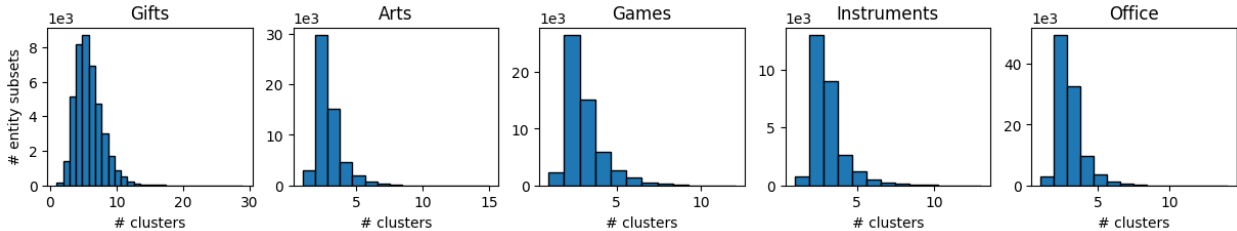

Figure A1: Histograms for number of clusters per entity subset for the five datasets described in Section 4.1.

## B   Self-Supervised Clustering

To improve the generalizability of CACTUS, especially when only a limited number of entity subsets with ground truth clusterings are available, we introduce a self-supervised clustering task. This task artificially creates ground truth clusterings using text augmentation techniques from contrastive learning. To create a training sample for self-supervision, we first randomly sample the number of clusters and the size of each cluster. Then, we randomly sample a seed entity (from the universal set of entities) for each cluster and populate each cluster with various random transformations of the seed entity. These transformations are obtained by randomly dropping words from the original description of the seed entity. The same augmented triplet loss function used for supervised clustering is used for self-supervised pretraining as well. The number of clusters is sampled from $\mathcal{U}(2, 10)$, cluster sizes are sampled from $\mathcal{U}(1, 5)$, and the fraction of words to drop from a seed entity is sampled from $\mathcal{U}(0.2, 0.7)$. We run self-supervised pretraining separately for each dataset using the universal set of entities specific to the corresponding dataset.

## C   Cross-Dataset Evaluation

Table A1 shows the performance of SCL, SCL (context), and CACTUS (ours) when the models are trained on one dataset and evaluated on another. See Section 4.2 for more details about the baselines. Both CACTUS and SCL (context) outperform SCL in most cases, indicating that the learned context-awareness is transferable across datasets. CACTUS typically performs better than SCL (context) for most source-target dataset pairs. However, there are specific cases where SCL (context) outperforms CACTUS, such

Table A1: Cross-dataset evaluation: Models are trained on a 'source' dataset and evaluated on the test set of 'target' dataset. (*Similar to Table 2 in the main paper, for the proprietary Gifts dataset, we report improvements against K-Means.)

| Source | Target | SCL | | | SCL (context) | | | CACTUS (ours) | | |
|---|---|---|---|---|---|---|---|---|---|---|
| | | AMI | ARI | F1 | AMI | ARI | F1 | AMI | ARI | F1 |
| Gifts* | Arts | 0.27 | 0.34 | 0.810 | 0.285 | 0.359 | **0.812** | **0.304** | **0.361** | 0.810 |
| | Games | 0.25 | 0.277 | **0.795** | 0.258 | 0.291 | **0.795** | **0.279** | 0.295 | 0.793 |
| | Instruments | 0.244 | 0.273 | 0.791 | **0.300** | **0.328** | **0.806** | 0.292 | 0.297 | 0.797 |
| | Office | 0.354 | 0.414 | 0.843 | 0.384 | **0.450** | **0.852** | **0.401** | 0.447 | 0.848 |
| Arts | Gifts | +0.03 | +0.035 | +0.052 | +0.084 | +0.059 | +0.021 | **+0.103** | **+0.101** | **+0.072** |
| | Games | 0.245 | 0.274 | 0.789 | 0.236 | 0.265 | 0.789 | **0.284** | **0.313** | **0.802** |
| | Instruments | 0.24 | 0.27 | 0.784 | 0.242 | 0.276 | 0.783 | **0.331** | **0.354** | **0.812** |
| | Office | 0.355 | 0.413 | 0.836 | 0.381 | 0.445 | 0.847 | **0.416** | **0.478** | **0.859** |
| Games | Gifts | +0.016 | +0.025 | **+0.051** | +0.051 | +0.028 | +0.001 | **+0.064** | **+0.063** | +0.046 |
| | Arts | 0.274 | 0.337 | **0.805** | 0.261 | 0.328 | 0.801 | **0.276** | **0.341** | 0.802 |
| | Instruments | 0.226 | 0.257 | 0.78 | 0.264 | 0.292 | 0.794 | **0.274** | **0.304** | **0.797** |
| | Office | 0.347 | 0.406 | 0.838 | **0.361** | **0.418** | **0.841** | 0.351 | 0.407 | 0.837 |
| Instruments | Gifts | +0.014 | +0.022 | +0.052 | +0.067 | +0.045 | +0.009 | **+0.087** | **+0.086** | **+0.069** |
| | Arts | 0.267 | 0.333 | 0.797 | 0.249 | 0.318 | 0.794 | **0.282** | **0.350** | **0.804** |
| | Games | 0.251 | 0.280 | 0.790 | 0.241 | 0.273 | 0.788 | **0.291** | **0.321** | **0.804** |
| | Office | 0.374 | 0.429 | 0.844 | 0.36 | 0.422 | 0.843 | **0.398** | **0.457** | **0.853** |
| Office | Gifts | +0.025 | +0.032 | +0.055 | **+0.070** | +0.045 | +0.01 | +0.069 | **+0.073** | **+0.066** |
| | Arts | 0.292 | 0.353 | 0.810 | 0.284 | 0.351 | 0.804 | **0.322** | **0.390** | **0.818** |
| | Games | 0.226 | 0.248 | 0.788 | 0.225 | 0.257 | 0.782 | **0.276** | **0.306** | **0.802** |
| | Instruments | 0.167 | 0.21 | 0.756 | 0.237 | 0.268 | 0.788 | **0.295** | **0.325** | **0.804** |

as when transferring from Games to Office and from Gifts to Office/Instruments. This suggests that while CACTUS is highly effective at leveraging contextual information for cross-dataset generalization, there are specific scenarios where SCL (context) may have an advantage in certain cases. Compared to unsupervised baselines in Table 2 (main paper), CACTUS achieves higher AMI, ARI, and F1 across all datasets, with the exception of the F1 score when trained on Gifts and evaluated on Games. These results highlight the importance of the proposed method in improving the robustness and adaptability of supervised clustering models across diverse datasets.

## D   Discussion of Clustering Algorithms

All the supervised clustering methods considered in Table 2 output pairwise cosine similarities between entity pairs within the given subset. The loss functions used to train these methods are also based on these similarities. During inference, these pairwise similarities are passed to the agglomerative clustering (average-link) algorithm. In this section, we provide a discussion on possible alternatives to agglomerative clustering.

We opted for agglomerative clustering because it works well with pairwise similarities and can output different numbers of clusters with varying sizes across different entity subsets. Prototype-based clustering methods, such as K-means and mixture models, are less suitable for our purpose since they require the number of clusters to be pre-specified. Density-based methods like DBSCAN may result in distant border points being included within the same cluster, which is also undesirable. The average-link agglomerative clustering algorithm used in this paper is a form of graph-based clustering. Single-link agglomerative clustering can also suffer from the same problem as density-based methods. Future work could explore the application of other graph-based clustering algorithms in our inference pipeline.

# E Implementation Details

We implemented the proposed methods and baselines in python using the HuggingFace Transformers library (Wolf et al., 2020). We adapted the T5ForConditionalGeneration class to implement the Scalable Inter-entity Attention (SIA) method. Our experiments were run on an Ubuntu 20.04.6 LTS server using a single NVIDIA Quadro RTX GPU. To ensure reproducibility, we seeded all random number generators before each experiment.

For both pretraining and finetuning, we used a learning rate of 1e-4 and a batch size of 4 during training, while a batch size of 16 was used for inference. Finetuning was run for 10 epochs, with evaluations on the validation set performed after each epoch. The checkpoint from the epoch with the best combined (sum of) NMI, AMI, RI, and ARI on the validation set was selected for evaluation on the test set. For all supervised methods, we ran average-link agglomerative clustering using predicted pairwise similarities, varying the threshold from -1 to 1 in increments of 0.1. The optimal threshold was selected based on performance on the validation set. Pretraining was run for 20K training batches.

For SCL baseline, we used the hyperparameter values of C and r as 0.15 and 0.5, respectively, as recommended in Haponchyk & Moschitti (2021). For both triplet loss and augmented triplet loss, we set the margin to 0.3, and initialized the neutral similarity $s_{neut}$ to 0 for augmented triplet loss.

For baselines that capture context by concatenating entity embeddings to the "average entity embedding in the subset" and then passing the concatenated representation to an MLP, we used an MLP with a single hidden layer. The input and hidden layers of this MLP contained $2d$ neurons each, the output layer contained $d$ neurons. The hidden layer employed 'tanh' as the activation function.

For unsupervised clustering methods, we computed the entity embeddings similarly to NIA but using the pretrained Flan-T5-base model without any finetuning. For K-Means and Spectral clustering, the number of clusters for each entity subset was determined using either the silhouette method or the average number from the training set, depending on validation metrics. For agglomerative clustering, we used cosine similarity with average linkage, setting the threshold based on the validation set. We used implementations of KMeans, Spectral, and Agglomerative clustering algorithms from scikit-learn (Pedregosa et al., 2011).

## E.1 Batch Processing

Below are the details of how we constructed batched inputs and obtained entity embeddings for the NIA, SIA, and FIA methods. Consider a batch where:

- $B$ = batch size (number of entity subsets in the batch)

- $K$ = maximum entity length (in number of tokens, which can be greater than the number of words)

- $L$ = maximum length when all entities in a subset are concatenated along with separator tokens

- $N$ = total number of entities in the batch (or sum of number of entities in all subsets in the batch)

- $M$ = maximum number of entities per subset

The input tensors for the three methods include

- FIA: A tensor for 'input_ids' of shape $B \times L$ containing token ids, a tensor for 'attention_mask' of the same shape to represent padded positions, and a tensor for 'entity_end_positions' of shape $B \times M$ which is used for identifying the end positions of entities in the concatenated lists.

- NIA and SIA: A tensor for 'input_ids' of shape $N \times K$ which contains token ids, a tensor for 'attention_mask' of the same shape, and a tensor for 'subset_entity_membership' of shape $B \times M$ which identifies the entities in each subset in the batch.

For FIA, we pass $B$ number of $L-$length padded sequences to the SLM. Here, each sequence is made up of the concatenation of all entities in a subset along with separator tokens. The hidden states from the last layer are averaged for each entity (using 'entity_end_positions') to obtain the entity embeddings. For NIA and SIA, we pass $N$ number of $K-$length padded sequences to the SLM. For SIA, the 'subset_entity_membership' tensor is used for inter-entity attention computations within each subset. All the three methods eventually generate a tensor of size $B \times M \times d$, which contains the $d-$dimensional entity embeddings for a maximum of $M$ entities in each of the $B$ subsets.

## F Metrics

This section defines the external clustering evaluation metrics used in the paper. Consider a ground truth clustering 'A' and a predicted clustering 'B' with the contingency matrix shown below.

|  | Clustering B | | | | |
| --- | --- | --- | --- | --- | --- |
|  | $n_{11}$ | $n_{12}$ | $\ldots$ | $n_{1s}$ | $a_1$ |
|  | $n_{21}$ | $n_{22}$ | $\ldots$ | $n_{2s}$ | $a_2$ |
| Clustering A | $\vdots$ | $\vdots$ | $\ddots$ | $\vdots$ | $\vdots$ |
|  | $n_{r1}$ | $n_{r2}$ | $\ldots$ | $n_{rs}$ | $a_r$ |
|  | $b_1$ | $b_2$ | $\ldots$ | $b_s$ | $n$ |

**Normalized Mutual Information**: $NMI = \frac{2\,MI(A,B)}{H(A)+H(B)} \in [0,1]$, where $H(A) = -\sum_{i=1}^{r} \frac{a_i}{n} log\big(\frac{a_i}{n}\big)$ and $H(B) = -\sum_{j=1}^{s} \frac{b_j}{n} log\big(\frac{b_i}{n}\big)$ are the entropies of clusterings 'A' and 'B', and $MI(A,B) = \sum_{i=1}^{r} \sum_{j=1}^{s} \frac{n_{ij}}{n} log\big(\frac{n_{ij}n}{a_i b_j}\big)$ is the mutual information between the two clusterings.

**Adjusted Mutual Information Vinh et al. (2009):** $AMI = \frac{MI - \mathbb{E}[MI(A,B)]}{\frac{H(A)+H(B)}{2} - \mathbb{E}[MI(A,B)]}$ adjusts MI for chance. Assuming the generalized hypergeometric distribution, the expected value of MI is given by $\mathbb{E}[MI(A,B)] = \sum_{i=1}^{r} \sum_{j=1}^{s} \sum_{n_{ij}=max(a_i+b_j-n,0)}^{min(a_i,b_j)} \frac{n_{ij}}{n} log\big(\frac{nn_{ij}}{a_i b_j}\big) \times \frac{a_i! b_j! (n-a_i)! (n-b_j)!}{n! n_{ij}! (a_i-n_{ij})! (b_j-n_{ij})! (n-a_i-b_j+n_{ij})!}$.

**Rand Index**: $RI = \frac{\alpha+\beta}{\binom{n}{2}} \in [0,1]$ where $\alpha$ is the number of pairs within same cluster in both A and B, and $\beta$ is the number of pairs in different clusters in both A and B.

**Adjusted Rand Index Hubert & Arabie (1985):** $ARI = \frac{RI - \mathbb{E}[RI]}{max[RI] - \mathbb{E}[RI]} \leq 1$ adjusts RI for chance. Assuming the generalized hypergeometric distribution, we can show that $ARI = \frac{\sum_{ij} \binom{n_{ij}}{2} - \big[\sum_i \binom{a_i}{2} \sum_j \binom{b_j}{2}\big]/\binom{n}{2}}{\frac{1}{2}\big[\sum_i \binom{a_i}{2} + \sum_j \binom{b_j}{2}\big] - \big[\sum_i \binom{a_i}{2} \sum_j \binom{b_j}{2}\big]/\binom{n}{2}}$. AMI and ARI are close to 0 for random clusterings.

**F1-score Haponchyk et al. (2018):** This is calculated as the harmonic mean of precision and recall, where $precision = \frac{\sum_j max(\{n_{1j},...,n_{rj}\})}{n}$ and $recall = \frac{\sum_i max(\{n_{i1},...,n_{is}\})}{n}$.

