# OpenReview forum: "Context-Aware Clustering using Large Language Models"
_TMLR — Rejected by TMLR_

### Review · Reviewer_pzcN · 2025-08-21

**Summary Of Contributions:**

The authors propose CACTUS -- a supervised clustering framework that distils clustering knowledge from closed-sourced LLMs into SLMs. The proposed method introduces three contributions: a) scalable inter-entity attention (SIA) mechanism to effectively capture contextual relationships among entities within subsets, b) augmented triplet loss with a learnable neutral similarity anchor to resolve margin inconsistencies in standard triplet loss and c) self-supervised clustering pretraining task leveraging text augmentation to improve generalization especially under limited labeled data. The authors conduct experiments on 5 datasets and show substantial gains over different baselines (both supervised and unsupervised) with improvements of 9-18% in AMI and ARI. Furthermore, the authors also conduct ablation studies to demonstrate the contribution of each component. Overall, the work offers a practical and scalable alternative to LLM-based clustering for real-world applications such as e-commerce, query suggestion and news aggregation.

**Audience:**

Yes

**Audience Explanation:**

I think this work would be useful to the TMLR audience and also quite relevant, as the authors explore the feasibility of SLMs for clustering.

**Claims And Evidence:**

Yes

**Claims Explanation:**

1. The authors use scalable inter-entity attention (SIA) that enables the model to incorporate subset-specific context efficiently overcoming a major limitation of existing clustering methods that often ignore inter-entity relationships.

2. The augmented triplet loss addresses inconsistencies of traditional triplet loss by introducing a neutral similarity anchor to ensure more stable training and better alignment across clusters. The results on validation show that it surpasses cross-entropy, structural loss and vanilla triplet on metrics like AMI, ARI and F1.

3. The authors also propose a clustering-specific pretraining task using text augmentation, which is an effective way to improve generalization, especially in low-resource settings where labeled data is scarce.

4. The authors evaluate CACTUS on 5 real-world datasets and show strong improvements between 9-18% across different metrics.

5. The authors also conduct detailed ablations to effectively disentangle the contributions of individual and combined contributions of SIA, augmented loss and self-supervised pretraining. The method is scalable with SLMs which makes it practically more feasible for real-world applications.

**Requested Changes:**

1. The ground-truth cluster annotations are currently collected using a closed-source LLMs and the experiments of this work are not reproducible. I would encourage the authors to also include experiments with open-source LLMs to ensure the results to be reproducible in addition to existing results.

2. The experiments mainly consider e-commerce datasets for this work. However, there are many other domains such as news articles, biomedical texts, academic papers where this work could also be applicable. I'm uncertain about the generalizability of the findings and the claim due to limited diversity of dataset.

3. The method uses agglomerative clustering at inference which requires careful threshold tuning. I think there is close to no discussion about the robustness of the results across the thresholds. It would be beneficial to quantify the sensitivity/variance across datasets due to the threshold.

4. The datasets used for the experiments include amazon product titles and gift queries which could be very short. It is unclear how the proposed method performs on longer and noisy descriptions. I think the noisy descriptions have been used as an augmentation technique by the authors for pretraining but would be extremely useful to quantify this in the current set of results.

5. The authors default to agglomerative inference for all supervised methods. There is a mention of alternative graph objectives/inference (e.g., correlation clustering or modern linkage/graph cuts) but the baselines with these objectives haven't been explored in the current work.

6. The authors should also report the actual runtime for the different baselines and the proposed method. I think currently only the memory requirement is reported for the inference.

---

### Review · Reviewer_VzJe · 2025-08-29

**Summary Of Contributions:**

Basically, what they’re doing with CACTUS is kind of like what we see when we make language models understand the context of a sentence. Except here, they’re applying that idea to clustering. So instead of just figuring out the meaning of a word based on the words around it, they’re figuring out how to group an item based on the other items in the same cluster. It’s like taking the whole “context-aware” idea we know from understanding text and using it to make clustering smarter. They’re doing that by tweaking the attention mechanism so the model can pay attention to relationships within a set of items. They also introduced this augmented triplet loss to make training more stable, and how they use a self-supervised pretraining step to help the model get better at clustering before fine-tuning. Overall, they’re blending the idea of context-awareness from language models into the world of clustering.

Weaknesses:
- Poorly structured introduction: The introduction does not clearly state research questions or hypotheses. It jumps straight into describing methods and jargon without building a logical progression (research question → hypotheses → contributions). As a result, the problem statement is vague and difficult to follow.
- Weak related work: The related work section is less than a page, very heavy in technical jargon, and omits many relevant and recent papers in supervised and deep clustering. It does not properly connect prior methods to the contributions of this paper. A stronger section would both broaden coverage and present the material with less technical density and better flow.
- Weak methodological motivation: The proposed method is not presented as a natural consequence of clear research questions. Similarly, the experimental design is not framed around testing specific hypotheses. This weakens the overall scientific narrative.
- Limited generality: The method is only evaluated on short-text clustering (product titles, queries). It is unclear whether the approach generalizes to longer documents or other domains.
- Missing comparisons: The experiments omit some recent and competitive deep learning methods for clustering, which makes it hard to fully situate the contribution relative to the current state of the art.

Strengths:
- Novel use of context-aware attention (SIA) for clustering short texts.
- Introduction of augmented triplet loss to stabilize training and improve cluster quality.
- Addition of a self-supervised pretraining step that boosts performance, especially in low-resource settings.
- Strong empirical results across multiple datasets with comprehensive ablations.

**Additional Comments:**

Overall, this paper introduces interesting and useful ideas for supervised clustering, particularly the scalable inter-entity attention mechanism and the augmented triplet loss. The empirical results are strong, with thorough ablations. However, the paper’s clarity and structure (especially in the introduction and related work) need significant improvement. The authors should also broaden the set of baselines, release code for reproducibility, and explicitly discuss generality beyond short texts. Addressing these issues would make the contribution much clearer and more impactful for the community.

**Audience:**

Yes

**Audience Explanation:**

Yes. The paper will interest TMLR’s audience because it addresses supervised clustering of short texts, a task relevant to NLP and e-commerce applications.

**Broader Impact Concerns:**

This work relies on cluster labels generated by a closed-source LLM (e.g., GPT-4) as ground truth. That raises concerns about transparency, reproducibility, and potential bias in the training data, since the quality and fairness of the clusters are dependent on the proprietary model’s outputs. These biases may propagate into CACTUS and affect downstream use cases such as e-commerce or search query clustering. In addition, the method is evaluated only on short-text data (products, queries), so its broader applicability and fairness in other domains are unclear. The authors should acknowledge these limitations explicitly and discuss potential societal impacts of deploying clustering systems trained on synthetic/LLM-generated supervision.

**Claims And Evidence:**

Yes

**Claims Explanation:**

Yes. The main claims are supported by the experiments and ablations provided. However, the credibility of the results would be stronger if (i) additional recent baselines were included in the comparisons, and (ii) the authors released code to allow reproducibility and independent verification.

**Requested Changes:**

- Improve the introduction:
-- Clearly state the research problem, research questions, and hypotheses.
-- Build a logical flow from problem → research question → hypotheses → contributions. Right now, this is missing and makes the paper hard to follow.
- Expand the related work section:
The related work needs to be significantly expanded (currently less than a page). Reduce jargon-heavy writing and improve the flow so it connects prior methods to your contributions. In particular, include recent supervised clustering works (e.g., NSC [Haponchyk & Moschitti 2021], SCL [Barnabò et al. 2023]), as well as other relevant strands: deep/semi-supervised clustering (e.g., Zhang et al. 2021/2022), set-encoding architectures (e.g., Set Transformer, Lee et al. 2019), and context-aware or LLM-based clustering approaches such as CafeLLM (Huang & Small 2023), Text Clustering with LLM Embeddings (Petukhova et al. 2023), and domain-specific context-aware clustering (e.g., Han et al. 2020, Liu et al. 2020). Discuss explicitly how CACTUS addresses the gaps these methods leave open.
- Missing baselines in experiments:
-- The experimental section should include comparisons with stronger and more recent baselines. In addition to K-means, Spectral, Agglomerative, Word-emb, and SCL, please evaluate against or at least discuss other competitive clustering approaches, such as NSC (if feasible on your hardware), contrastive/deep clustering models, and recent LLM-based clustering methods (e.g., CafeLLM, Few-Shot Clustering with LLMs). This is important for situating CACTUS relative to the true current state of the art.
- Code availability:
-- Provide a link to a GitHub repository with your implementation to allow reproducibility and independent verification of results. Code release is critical for credibility.
- Clarity of methodology presentation:
-- Connect the design of the method back to the research questions (e.g., why the augmented triplet loss addresses the stated problem, how SIA emerges from the hypothesis about context).
Improve explanation of SIA with more intuitive examples or diagrams.
-- Discuss potential performance and applicability on longer texts (e.g., news articles, documents) and not only short product titles or queries.
-- If possible, include at least one experiment on a dataset beyond e-commerce to demonstrate broader applicability.
- Experiment design linkage:
-- Clearly state how each experiment corresponds to testing a hypothesis or research claim (e.g., ablations test “context helps,” loss comparisons test “augmented triplet stabilizes learning”).
- Writing and structure:
-- Reduce overly technical jargon in the introduction and related work so that the narrative is accessible to a broader TMLR audience.
-- Ensure smoother transitions between sections and highlight contributions at the end of the introduction.

---

### Review · Reviewer_BfCb · 2025-08-31

**Summary Of Contributions:**

The authors of the paper address the challenge of using LLMs for text clustering tasks. They observe that despite the high-quality results produced by LLMs, their massive size and cost make them impractical for real-world use.

To solve this, the researchers propose a systematic approach called **CACTUS**, which transfers clustering knowledge from LLMs to smaller models. The method is designed for supervised clustering and introduces several innovations:
* It incorporates a scalable inter-entity attention mechanism to model the relationships between text items within a subset, something that existing methods typically fail to do.
* It proposes a new augmented triplet loss function, which is specifically tailored to the challenges of supervised clustering.
* It includes a self-supervised pretraining task to improve the model's ability to generalize

**Additional Comments:**

n/a

**Audience:**

Yes

**Audience Explanation:**

The problem is well motivated and might be of practical relevance.

**Claims And Evidence:**

Yes

**Claims Explanation:**

I am not an expert in the field. I was able to make a rather superficial assessment of the paper. In my view:
1. The problem is well-motivated. I am not completely convinced about the practicality, though. While, I understand that using smaller models is beneficial, I'd love to see some analysis, at what scale the effort of training and deploying a specialised models pays off.
2. The proposed method seems sound. The choice of the particular loss is not entirely clear for me (e.g. why not some other contrastive loss).
3. The experimental side feels sufficient.
3a. Unfortunately, I cannot fully asses the strength of the method as I am not able to interpret the metrics.
3b. The 'number' of eval dataset is ok. I'd love to know if they are diverse enough. What concerns me, for example, is if they cover various sizes of clusters, and granularity. The information in the paragraph 2 p. 8 is somewhat rudimentary.
3c. I'd love to see more of information about how (the size and quality of dataset, the size of the model) affect the quality.

**Requested Changes:**

Please see the comments in above. In particular, I'd love to see some discussion about the practicality of the proposed solution. For example, what is the minimal volume of traffic at which the method would be cheaper, if we take into account all the cost (including .e.g training).

---

### Decision · Action_Editor_7Lp9 · 2025-10-07

**Recommendation:** Reject

**Additional Comments:**

The paper claims to offer a "scalable, generalizable solution for real-world clustering scenarios". However, the empirical validation is confined to a single, narrow domain of short-text e-commerce datasets. This represents a foundational mismatch between the paper's broad claims and its supporting evidence, which is the primary reason for this rejection under TMLR's core criterion of evidential support.

Overall, the work as presented might be better suited for a specialized venue focused on applied NLP or e-commerce, where the narrow experimental focus could be considered a strength. AE suggests that the author submit to those specialized venues.

**Audience:**

Yes

**Audience Explanation:**

The reviewers acknowledged that the work is well-motivated and the ideas are interesting. The core challenge, transferring clustering capabilities from large, expensive LLMs to smaller, more efficient models, is both timely and practical.

**Claims And Evidence:**

No

**Claims Explanation:**

Several critical concerns undermine the current submission's technical soundness and the strength of its empirical evaluation:

1. Limited Scope of Evaluation: The experiments are confined to short-text e-commerce datasets (product titles and queries). This narrow focus does not adequately support the broad claim that CACTUS is a "scalable, generalizable solution for real-world clustering scenarios". Reviewers correctly pointed out the need to evaluate the method on more diverse domains (e.g., news, biomedical) and on longer, noisier documents to establish generalizability.

2. Missing State-of-the-Art Baselines: The paper fails to compare against several recent and highly relevant baselines in deep, contrastive, and LLM-based clustering. Without these comparisons, the claim that the proposed approach "significantly outperforms existing... baselines"  is not fully substantiated. To be convincing, the evidence must situate the work against the true state of the art and those been widely used in real-world practices, such as NV-embed (https://arxiv.org/abs/2405.17428).

3. Reproducibility and Transparency Concerns: The work relies on a closed-source, proprietary LLM to generate the ground-truth labels, raising significant concerns about reproducibility, transparency, and potential label bias.